# Modeling of Nitrogen Removal from Natural Gas in Rotating Packed Bed Using Artificial Neural Networks

**DOI:** 10.3390/molecules28145333

**Published:** 2023-07-11

**Authors:** Amiza Surmi, Azmi Mohd Shariff, Serene Sow Mun Lock

**Affiliations:** 1Chemical Engineering Department, Universiti Teknologi PETRONAS, Bandar Seri Iskandar 32610, Perak, Malaysia; amiza_surmi@petronas.com; 2Group Research & Technology, Petroliam Nasional Berhad (PETRONAS), Lot 3288 & 3289, off Jalan Ayer Itam, Kawasan Institusi Bangi, Kajang 43000, Selangor Darul Ehsan, Malaysia; 3Institute of Contaminant Management, CO2 Research Centre (CO2RES), Universiti Teknologi PETRONAS, Bandar Seri Iskandar 32610, Perak, Malaysia; sowmun.lock@utp.edu.my

**Keywords:** artificial neural networks, carbon dioxide, nitrogen, liquefied natural gas

## Abstract

Novel or unconventional technologies are critical to providing cost-competitive natural gas supplies to meet rising demands and provide more opportunities to develop low-quality gas fields with high contaminants, including high carbon dioxide (CO_2_) fields. High nitrogen concentrations that reduce the heating value of gaseous products are typically associated with high CO_2_ fields. Consequently, removing nitrogen is essential for meeting customers’ requirements. The intensification approach with a rotating packed bed (RPB) demonstrated considerable potential to remove nitrogen from natural gas under cryogenic conditions. Moreover, the process significantly reduces the equipment size compared to the conventional distillation column, thus making it more economical. The prediction model developed in this study employed artificial neural networks (ANN) based on data from in-house experiments due to a lack of available data. The ANN model is preferred as it offers easy processing of large amounts of data, even for more complex processes, compared to developing the first principal mathematical model, which requires numerous assumptions and might be associated with lumped components in the kinetic model. Backpropagation algorithms for ANN Lavenberg–Marquardt (LM), scaled conjugate gradient (SCG), and Bayesian regularisation (BR) were also utilised. Resultantly, the LM produced the best model for predicting nitrogen removal from natural gas compared to other ANN models with a layer size of nine, with a 99.56% regression (R^2^) and 0.0128 mean standard error (MSE).

## 1. Introduction

In Malaysia, over 13 trillion standard cubic feet per day (Tscfd) of undeveloped gas fields have been identified [1]. Commonly, high carbon dioxide (CO_2_) fields are associated with high nitrogen concentrations, which makes the treatment of natural gas more challenging. Nevertheless, removing CO_2_ to meet the 6.5-mole percentage (mol%) or less pipeline specification might result in greater nitrogen concentrations than what customers require. Since nitrogen is inert, removing the gas maximises the calorific value of the product gas while minimising any safety issues, particularly during liquefied natural gas (LNG) shipping due to stratification and rollover of the product [2]. Consequently, most LNG production plants limit nitrogen to a maximum of 1.0 mol% in their LNG outputs. Nonetheless, removing or venting nitrogen into the atmosphere necessitates a small amount of methane (typically under 1%), which could present a safety hazard as methane is a combustible gas. Methane is also a greenhouse gas that requires control during venting or flaring [3].

In a study [2], it was found that only cryogenic technologies could remove nitrogen to under 1.0 mol% and produce liquefied methane at large plant capacities. Meanwhile, non-cryogenic technologies, such as absorption and adsorption, reportedly remove nitrogen at higher product specifications for feed rates under 15 million standard cubic feet per day (MMSCFD). Consequently, gas separation utilising membranes has been extensively studied and applied. Nonetheless, unlike CO_2_ removal from natural gas, nitrogen removal with membranes is reportedly minimal due to the inefficient methane and nitrogen separation to meet the under 1 mol% due to their similar kinetic sizes, 0.36 nm (nitrogen) and 0.38 nm (methane) [2,4]. 

A general nitrogen removal and liquefaction process employs a flash vessel to liquefy natural gas with methane to meet the LNG nitrogen requirements. Nonetheless, utilising the off-gas from the flash vessel as fuel gas results in hydrocarbon and product losses. Consequently, the cryogenic distillation concept maximises the hydrocarbon recovery from the off-gas. The tall distillation column and larger equipment required for the technique render it unattractive to offshore and onshore plants with limited plant size. Thus, process intensification (PI) is more appealing given its potential to overcome the challenges of maintaining dominance as the primary energy supplier while simultaneously attaining environmentally friendly and sustainable evolutions in the industry [5].

The PI utilises novel concepts and principles that improve chemical industry processes for sustainable product manufacturing. This method reportedly dramatically increased mass and transfer, reduced volume, equipment size and footprint, and operational and capital costs, allowed more sustainable material applications, simplified processes, and offered safer operations [5,6,7,8,9,10,11,12,13,14,15,16]. In the late 1970s, Professor Colin Ramshaw introduced PI to Imperial Chemical Industries (ICI). He employed a high gravitational force, Higee, to improve the mass transfer in the separation stage, intensify the process, and reduce the equipment size [7,8,16]. The technology utilised a novel rotating device to improve the gravitational force by over 100 times. The advantages of the Higee technology include intensified mass transfer with a very thin liquid film that attracts polymerisation, absorption, synthesis, conversion, and distillation [17]. 

High-gravity or centrifugal technologies include spinning disk reactor [11,18,19,20], static mixer [21,22,23], agitated slurry reactor [24], rotating zig-zag bed [25,26,27,28], and rotating packed bed (RPB). Guo et al. (2019) suggested over five RPB commercial applications in the industry, including desulphurisation, denitrification, particulate removal, and emission control. The approach offered equipment size, performance, and cost advantages over conventional technologies [29]. Furthermore, Boodhoo and Harle (2013) discovered that the RPB provided the highest mass and heat transfer compared to other PI technologies with significant equipment size and footprint reductions that benefit pilots or commercial applications [30]

Over the last few years, numerous experiments [31,32,33,34], simulations and modelling [35,36,37,38], and computational fluid dynamics (CFD)-based hydrodynamic studies [39,40,41,42] were conducted to procure a better understanding of the mass transfer within the RPB in high gravity environments and to improve the technology. Nonetheless, a majority of the reports were based on the absorption process and were operated under five bars and above zero temperatures. Limited studies also focused on RPB distillation, which included vacuum distillation and alcohol and heptane–hexane separations. No investigations have been conducted on nitrogen removal from natural gas using RPB distillation under cryogenic conditions and high pressures. 

Reports focusing on RPB optimisation via process simulations and modelling, such as CFD and mathematical modelling, are available. Nonetheless, CFD and mathematical modelling are complex and require significant input conditions to obtain accurate prediction models. Consequently, the artificial neural network (ANN) model is preferred due to its simplicity and ability to process massive amounts of data, even for more complex processes, compared to developing first principal mathematical models, which require numerous assumptions and might be associated with lumped components in kinetic models [43]. Furthermore, ANN does not require a more complex mathematical understanding to interpret data procured for analysis; hence, it is more suitable for process optimisation in several engineering fields [44,45,46,47].

The ANN is an example of machine learning that employs a nonlinear regression algorithm approach. The model is designed based on the behaviours of the human brain, thus requiring experience and training for more accurate prediction via interconnected neurons. The ANN consists of an input component that receives external signals and data, an output unit that outputs the system processing results, and a hidden segment that is not observable outside the system, situated between the input and output pieces [48].

The ANN is widely employed in chemical engineering for thermodynamic application studies, process design, control, optimisation, safety, experimental data fitting, and machine learning [46,47,49,50]. Moreover, research interest in machine learning in PI for chemical and process engineering has increased exponentially by 26% from 2015 to 2020 [51]. For example, Popoola and Susu (2014) utilised ANN with multiple inputs and outputs to determine the temperature cut-off points of kerosene, diesel, and naphtha products [52]. The input in the study was employed to design a crude distillation column control. 

Controlling the temperature of distillation columns is especially crucial in the industry. The process is very dynamic due to incoming process feed condition uncertainties, hence requiring a more robust control to maintain the pressure and temperature and achieve a good mass and energy balance to meet product specifications [49,53]. Wang and co-workers applied a neural network in the temperature proportional–integral–derivative (PID) controller design of a distillation column, considering that neural networks possess better adaptive and fault tolerance abilities [54]. Moreover, [47,49,55,56,57] reported a successful ANN implementation in distillation controls. Nevertheless, only a few applications are reported on RPB related to ANN model research, while none are available on studies or applications of ANN for cryogenic nitrogen removal utilising RPB. 

Saha (2009) developed a prediction model with ANN radial basis function to predict the volumetric gas side mass transfer coefficient based on experimental data from previous studies [58]. The report discovered an improvement of up to 15% with better accuracy than the empirical equations employing experimental data [58]. In another study, Lashkarbolooki et al. (2012) investigated pressure reductions in RPB equipment. The pressure gradient in an RPB is critical to allow counter-current contacts between the gas and liquid, particularly within the rotor or packing, to achieve a better mass transfer. The study employed an ANN model comprising 14 hidden layers and observed an excellent agreement between the model and experiments, with a 5.27% AARD, 3.0 × 10^−5^ mean square error (MSE), and 0.9985 regression (R^2^) [59]. Some reports focused on investigating contaminant removal efficiency from process gas using RPB technology for CO_2_ capture [60], adsorption [61], dust removal [62], and ozonation [63].

The present study aimed to develop ANN prediction models for nitrogen removal from natural gas based on in-house experimental data with various process parameters as inputs to meet the nitrogen removal efficiency output product. This study employed three artificial neural network training algorithms, with ANN models utilised for a given multilayer perceptron (MLP) feedforward neural network: the Lavenberg–Marquardt (LM), Bayesian regularisation (BR), and scaled conjugate gradient (SCG). The LM and BR training algorithms were based on the backpropagation algorithm. The swift convergence of LM is its primary benefit [64], whereas BR offers less probability of it being overfitted [65]. 

## 2. Results and Discussion

### 2.1. The Influences of High Gravity Factor on Removal Efficiency

A high gravity factor is essential in an RPB, which generates a high centrifugal acceleration to facilitate an effective mass transfer. The high gravity factor is determined based on the RPB radius and its rotational speed (see Equation (1)). The centrifugal acceleration resulting from high gravity is a key factor distinguishing RPB from conventional packed columns. Only eight studies have been conducted on ANN for RPB to date. All of the reports utilised a high gravity factor or rotational speed as their ANN modelling input for various applications and achieved over 94% R^2^ and under 1% MSE (see Table 1).
(1)$$\beta =\frac{2\omega^{2}\left(r_{1}^{2}{+r}_{1}r_{2}+r_{2}^{2}\right)}{3\left(r_{1}+r_{2}\right)g}$$
where r_1_ and r_2_ are the inner and outer radii of the rotor (m), respectively, and ω denotes the angular velocity in the RPB (rad/s). 

The high gravity factor in the current study was altered between 10 and 90, while the parameters were maintained throughout the experiment to investigate the effects of high gravity on nitrogen removal efficiency. As shown in Figure 1, an improved nitrogen removal efficiency was observed due to the increased high gravity factor. These findings could result from better contact between the gas and liquid at higher centrifugal accelerations, thereby contributing to better contaminant removal [66]. Similar effects were reported for other contaminant removal applications, such as hydrogen sulphide (H_2_S) [67], oxygen [38,68], and volatile organic compounds (VOC) [69,70].

Figure 2 illustrates the effects of the rotational speed and operating pressure on the nitrogen removal efficiency. Increasing the rotational speed enhanced the nitrogen removal efficiency, which decreased when the rotational speed reached 800 rpm. The diminished performance might be due to the reduced contact time between the gas and liquid. Conversely, at 200 rpm, the nitrogen removal performance began to decrease with increasing operating pressure. Nonetheless, the removal efficiency improved when the rotational speed was increased to 500 rpm and demonstrated a more stable condition, even at operating pressure variations. The results of this study are comparable to the findings on the prediction of vacuum distillation performance in the RPB reported by Li et al. (2017) [35].
molecules-28-05333-t001_Table 1Table 1The RPB modelling with ANN.AuthorApplicationsANN ModellingInput ParametersOutputParameterResultsWang et al., 2022[63]Degradation of bisphenol A (BPA) ozonation-Concentration-pH-Flowrate-Gravity factorBPA degradation efficiencyR^2^ = 0.9827MSE = 0.0003305Li, 2021[70]Volatile organic compound removal-Gravity factor-Reynold-Concentration-Henry’s constantVOC removal efficiencyR^2^ = 0.9697MSE = 0.0364Wei et al., 2018[71]Biosorption process absorption using agricultural waste-Gravity factor-Reynold-Contact time-Particle size-ConcentrationBiosorption timeR^2^ = 0.996MSE = 0.0000904 Li et al., 2017[62]Dust removal via absorption process-Reynold (gas, liquid, rotational)-Particle sizeSeparation efficiencyR^2^ = 0.9952MSE = 0.00013Li et al., 2016[72]Wastewater treatment using adsorption process-Gravity factor-Reynold number-Contact time-ConcentrationAdsorption efficiencyR^2^ = 0.9965MSE = 0.00016Zhao et al., 2014[60]CO_2_ capture in RPB using absorption process-Reynold-Schmidt-Grashof-Diffusion and mass transferCO_2_ capture efficiencyR^2^ = 0.9457MSE = 0.0012Lashkarbolooki et al., 2012[59]Prediction of pressure drop-Reynold (gas, liquid, and rotational)Pressure dropsR^2^ = 0.9985MSE = 0.00003Saha,2009[58]Mass transfer coefficient prediction-Liquid velocity-Gas velocity-rotationalMTCNot reported


Singh et al. (1992) proposed the area transfer unit–number of transfer unit (ATU-NTU) concept to explain the changes in fluid loading along the annular packing radius of an RPB [73]. The NTU is one of the most critical parameters to consider when assessing separation targets for distillation. Generally, determining the number of transfer units (NTU) involves evaluating the integrals in the equation describing the rectification and stripping sections, as shown in Equations (5)–(7). An increased NTU is anticipated if the performance target is set too low or if a thorough contaminant removal is attempted, which could result in an extremely tall column for conventional distillation. On the other hand, high centrifugal acceleration during separation leads to shorter residence times and process intensification, thereby requiring smaller equipment.

Figure 3 depicts ATU variations at different rotating speeds. According to the results, lower ATU values were documented at high operating speeds; hence, a shorter column was required to meet the product specifications. These observations are consistent with the findings reported by Qammar et al. (2018) on ethanol–water separation via total reflux distillation [34].
(2)$$\pi \left(r_{o}^{2}-r_{i}^{2}\right)=\left({ATU}_{G}\right)\left({NTU}_{G}\right)$$
(3)$${ATU}_{G}=\frac{G}{\rho_{G}{hK}_{G}a}$$
(4)$${NTU}_{G}=\int_{y2}^{y1}\frac{dy}{y*-y}$$

### 2.2. Comparisons between the ANN Models

The neural network architecture correlates with the inputs, hidden layer numbers, and neuron transfer functions [74]. The hidden layers in the current study were varied between 5 and 15 to determine the LM, SCG, and BR prediction models with the highest accuracy. At layer nine, the LM model recorded higher R^2^ and lower MSE values of 99.56% and 0.0128, respectively (see Figure 4). Nevertheless, increasing the hidden layer to more than nine led to more prominent MSE figures and a slightly reduced R^2^. 

The SCG model employed in the present study produced MSE values under 0.2%. Nonetheless, the 5–10 hidden layers in the model documented a good R^2^ trend and a slightly diminished R^2^ during validation and assessment with an increasing number of hidden layers (see Figure 5). Figure 6 demonstrates the relatively consistent R^2^ and MSE for training and validation of the BR model but documented some reduction during evaluations. At nine layers, the R^2^ and MSE of the BR model were 98.89 and 0.0493%, respectively, which were slightly better than those of the SCG by 0.0072%. 

The modelling results in this study revealed that increasing the number of layers resulted in higher MSE and lower R^2^ values, especially for the LM model. These findings could be due to overfitting. The R^2^ results were better when fewer hidden layers (6–10) were utilised. The LM model architecture with nine hidden layers was the optimal structure, which also produced a superior prediction than the SCG and BR models. Nonetheless, the accuracy of the model might be compromised if the number of hidden layers is too low, thus reducing the probability of meeting the target figures. Consequently, training, evaluating, and validating the models to ascertain the optimal number of hidden layers are necessary. Figure 7 illustrates the outcomes of the LM model based on the predicted model and experimental data. Based on the results, R^2^ yielded 96.66% of the 15% test data; therefore, it can be utilised to predict nitrogen removal from natural gas in the future.

## 3. Methodology

### 3.1. Experimental Setup

The primary components of the RPB utilised in the experimental setup for the cryogenic distillation-RPB (CD-RPB) process to remove nitrogen from natural gas in the current study (see Figure 8). Natural gas with a nitrogen concentration of up to 20 mol% was introduced to the CD-RPB via a liquid inlet post-chilled to ~−120 °C. A liquid distributor was designed for the inlet liquid to procure the spray effect, forming tiny droplets and a thin liquid film on the surface of the packing inside the CD-RPB. 

The inlet and reflux liquids from the reflux vessel and vapour flow from the reboiler utilised in this study were contacted in a counter-current manner, thereby providing improved mass transfer and nitrogen removal efficiency. The rich nitrogen was discharged as a gas phase, whereas the main product, LNG, was obtained as the bottom product at −161 °C. In the present study, the CD-RPB motor was installed at the bottom of the pressure vessel. The rotational speed was varied from 100 to 1000 rpm during the experiments. 

The RPB was equipped with stationary housing, liquid distributors, inner and outer diameters, packing or rotor, and a motor. The low-temperature and high-pressure evaluations (12–15 bars) employed in this study resulted in a quite complex cryogenic experimental setup. Consequently, other auxiliary equipment was crucial to support the CD-RPB setup and assessment. Equation (5) was employed to determine the nitrogen removal efficiency (ռ).
(5)$$ռ=\frac{C_{i}-C_{o}}{C_{i}}\times 100\%$$
where ռ is the nitrogen removal efficiency, C_i_ denotes the inlet concentration, and C_o_ represents the outlet concentration.

The setup used in this study was intended to process 840 kg of feed gas per day. A methane and nitrogen gas mixture was employed as the feed gas. The gas mixture was fed into the mol-sieve vessel to remove up to 1.0 part per million (ppm) moisture. A Shaw metre inline analyser was installed to monitor the quality of the gas, ensuring that it met the water specifications. The data were critical for low-temperature processes to avoid downstream processes of hydrate formation before sending the gas to the chiller and CD-RPB. Subsequently, distillation separated the nitrogen from methane based on its relative volatility. The rich nitrogen was then removed as the top product, while the bottom yield was LNG. The products were analysed using gas chromatography (GC) with a direct online sampling line connection. Furthermore, the evaluation setup included base layer process control, such as chiller temperature control and CD-RPB pressure control, to maintain the stability of the process.

### 3.2. Model Development with Neural Networks

The ANN model in this study possessed a minimum of three layers: input, hidden, and output. Figure 9 demonstrates the complex interconnection between the layers to produce the outputs [46,66]. The most complex architecture in the ANN was the ‘black box’, which is the hidden layer connecting the input and output variables. The structure is known as the ‘black box’ due to the still unknown technical explanation within the hidden layer.

The number of hidden layers depends on the architecture of the ANN model to achieve an acceptable r-square (R^2^) and lower means square error (MSE), as described in Equations (6) and (7). The input variables employed in the current study were the process parameters that were adjusted and controlled during the experiments to obtain the targeted values. The parameters and specifications of the model utilised in this study are summarised in Table 2. 

The obtained MSE and regression values, R, reflected the quality of the results in the present study. The average squared variations between the outputs generated by the MATLAB function and the targets (true, measured data corresponding to the inputs provided to the MATLAB functions) are denoted by the MSE. Consequently, since smaller figures were preferred, the algorithm with minimal MSE was considered the most suitable. The MSE was calculated using Equation (6). R correlated the obtained ANN outputs and the targets. An R value of 1 corresponded to a close association, while a value of 0 represented a random link. The R value was obtained by utilising Equation (7). The t value represented the arithmetic mean of the target values.
(6)$$MSE=\frac{1}{N}\sum_{i=1}^{N}{\left(e_{i}\right)}^{2}=\frac{1}{N}\sum_{i=1}^{N}{\left(t_{i}-a_{i}\right)}^{2}$$
(7)$$R^{2}=1-\frac{\sum_{i=1}^{N}{\left(t_{i}-a_{i}\right)}^{2}}{\sum_{i=1}^{N}{\left(\overset{-}{t_{i}}-t_{i}\right)}^{2}}$$
where N represents the sample numbers (input–output pairs) utilised for training the network, and t denotes the target value. 

Figure 10 depicts the conceptual structure and the ANN modelling approach utilised in this study. The data collection process was initiated with experimental procedures and continued with data screening and input categorisation. The input data were then fed into the ANN models for training, validations, and testing. Data from the experiment were employed for all consistent ANN models. Inconsistencies were observed when variables, such as R^2^ and MSE, were recorded on the hidden layers. 

All the ANN models in the current study followed an identical set of instructions. After data collection and examination, several models were considered to determine which provided the best and most accurate predictions. A MATLAB (R2022a) equipped with a version 8.3 toolbox was employed to perform the evaluations.

## 4. Conclusions

This study established three types of ANN models for nitrogen removal from natural gas utilising an RPB based on experimental data. The high gravity factor is a critical condition in RPB applications as it promotes vapour–liquid contact and enhances fluid mass transfers during contaminant removal. Consequently, the high gravity factor in the current study was varied from 30 to 90 to maintain a nitrogen removal efficiency above 97%.

Other process inputs, such as pressure, temperature, and concentration, are also essential for producing a good prediction model to remove nitrogen from natural gas under cryogenic operating conditions. The LM model performed ideally with nine hidden layers, with 99.56% R^2^ and 0.0128 MSE compared to the SCG and BY models. Nevertheless, all models produced over 90% R^2^ at different hidden layers. 

The increased number of layers almost consistently reduced the R^2^ and enhanced the MSE across all three ANN models. These findings might be due to data overfitting to meet the output. Nonetheless, the correlation inside the hidden layers remained unknown, hence considered a ‘black box.’ The results of this study suggested that the LM model was the best model for predicting nitrogen removal efficiency when employing an RPB under cryogenic conditions. Furthermore, reliance on the experimental setup and requirements was minimised. Consequently, the predicted model was reliable and applicable to process optimisations, techno-economic studies, and technology upscaling for commercialisation. 

## Figures and Tables

**Figure 1 molecules-28-05333-f001:**
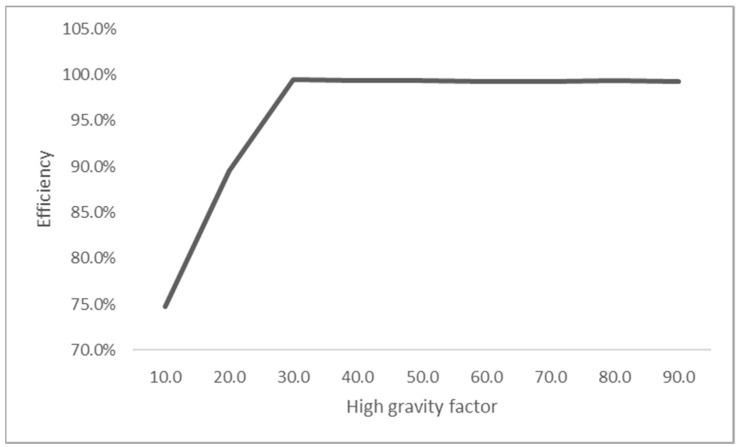
The effects of high gravity on nitrogen removal.

**Figure 2 molecules-28-05333-f002:**
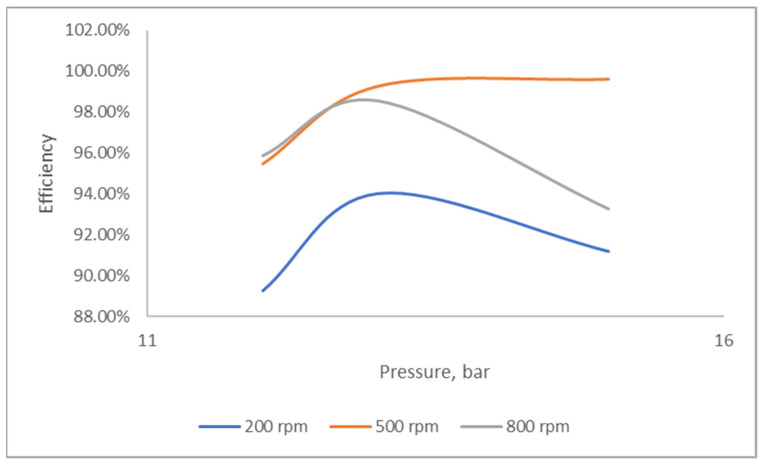
Effects of rotating speed on nitrogen removal at different pressures.

**Figure 3 molecules-28-05333-f003:**
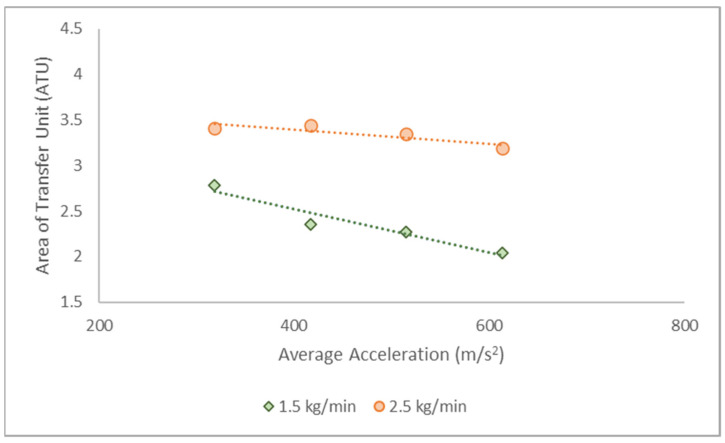
The ATU at varying average accelerations and flow rates.

**Figure 4 molecules-28-05333-f004:**
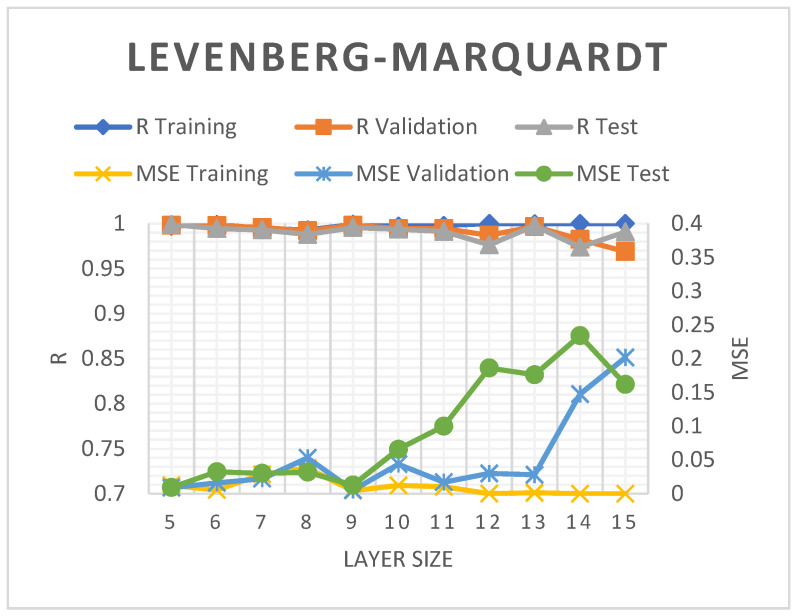
The R^2^ and MSE of the LM-ANN model.

**Figure 5 molecules-28-05333-f005:**
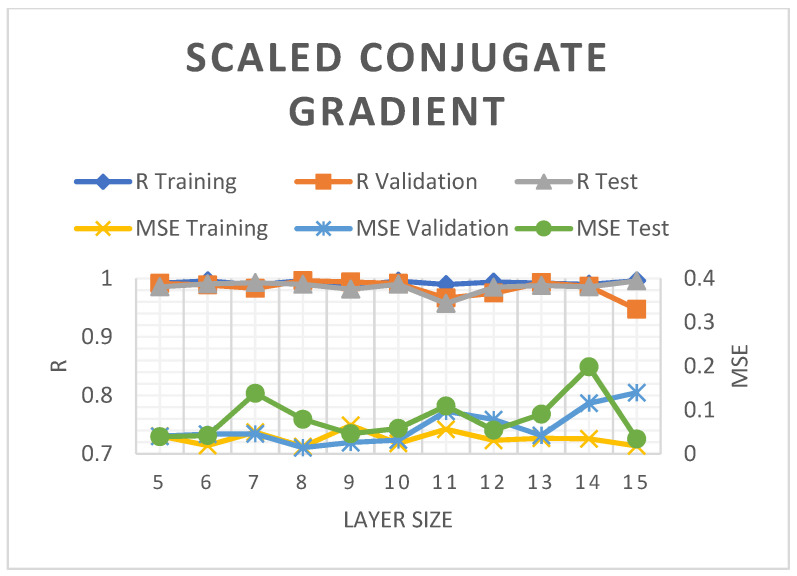
The R^2^ and MSE of the SCG-ANN model.

**Figure 6 molecules-28-05333-f006:**
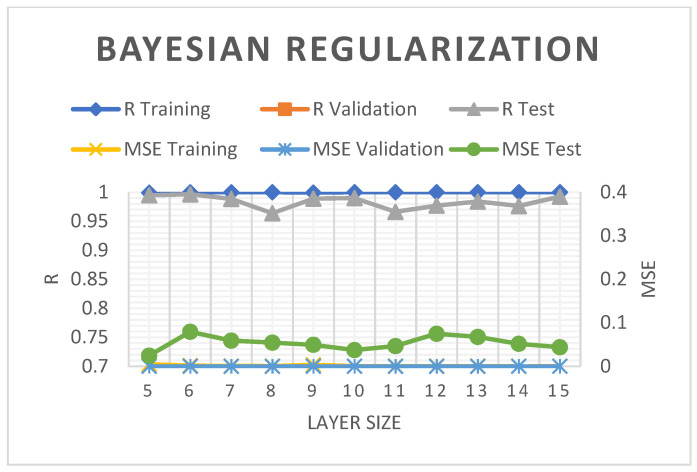
The R^2^ and MSE of the BR-ANN model.

**Figure 7 molecules-28-05333-f007:**
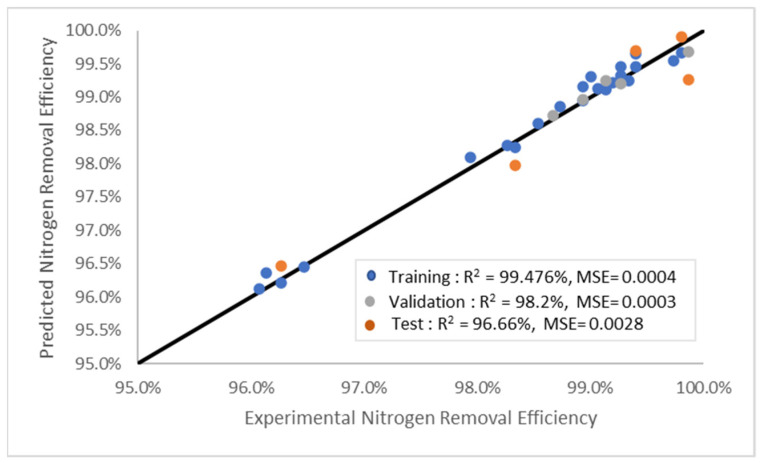
Nitrogen removal efficiencies of the experimental and prediction models.

**Figure 8 molecules-28-05333-f008:**
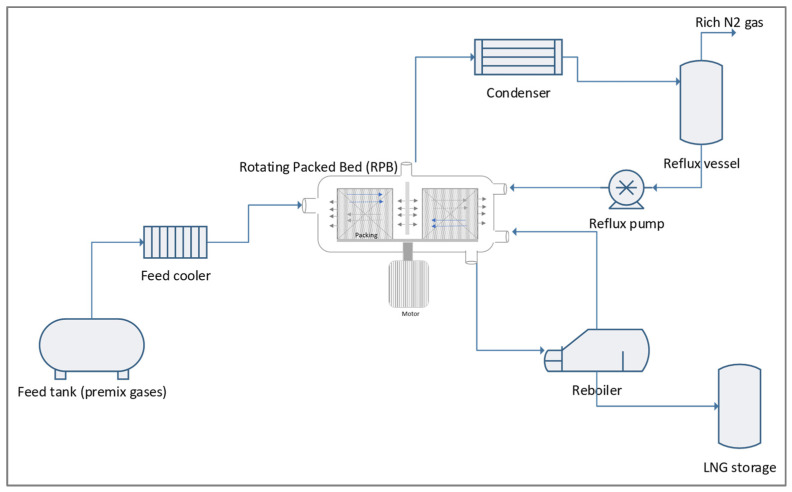
Simplified block diagram of the cryogenic nitrogen removal system experimental setup.

**Figure 9 molecules-28-05333-f009:**
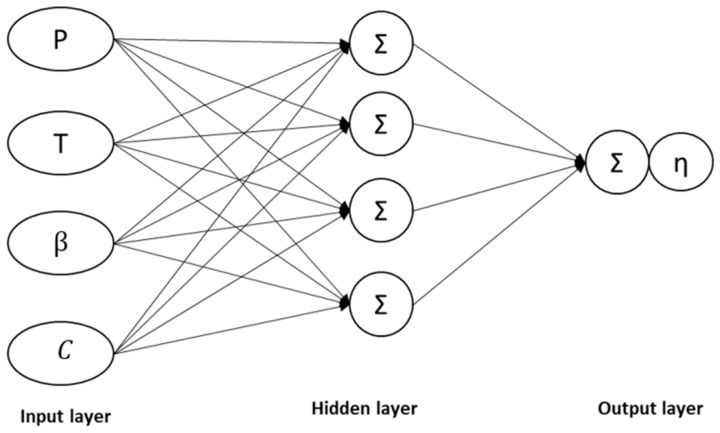
The typical neural network architecture.

**Figure 10 molecules-28-05333-f010:**
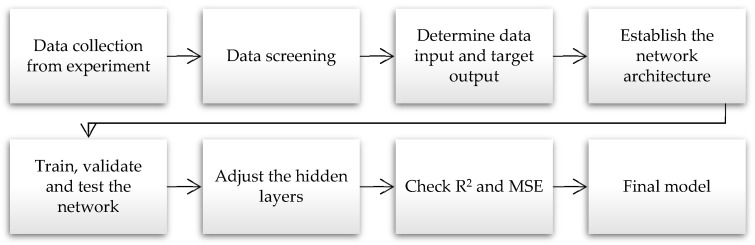
The ANN modelling flowchart.

**Table 2 molecules-28-05333-t002:** The parameters and specifications of the models.

Parameters	Model Specifications
Model	Levenberg–Marquardt (LM) Bayesian Regularization (BR) Scaled Conjugate Gradient (SCG)
Samples’ distribution [63]	Training: 75% Test: 15% Validation: 15%
Number of inputs	4
Number of outputs	1
Hidden layer transfer function	Sigmoid
Output layer transfer function	Linear
Number of data sets	45

## Data Availability

Not applicable.

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
