# Peer review of "Modeling of Nitrogen Removal from Natural Gas in Rotating Packed Bed Using Artificial Neural Networks"

_molecules, 2023, doi:10.3390/molecules28145333_

Round 1

Reviewer 1 Report

Removing the impurities in nature gas can raise its grade. The manuscpript developed a prediction model employed artificial neural networks (ANN) therory based on data from in-house experiments. Due to lacking available data, it is a plain paper with limited value. Although, I think it  might be accepted after some revision.

1. It only displays the influence of gravity in figure 4, how about some other factors? The authors also said that "Other process inputs, such as pressure,temperature, and concentration are also essential" in the conclusion part

2. the efficiency is  in the range of 99%-99.5%, What's the point of such a small range of fluctuations? Can the gravity be some littler? eg, 5 or 10?

Language shall be polished. For example, the subject of the sentence P1L44 is missing. Some subheadings formula is not correct.

Author Response

Dear Reviewer,

Many thanks for reviewing this manuscript. The comments are addressed in the attachment.

Thank you

Reviewer 2 Report

In this paper, three types of ANN models was established for nitrogen removal from natural gas utilising an RPB based on experimental data. The high gravity factor is a critical condition in RPB applications as it promotes the vapour liquid contact and enhances fluid mass transfers during contaminants removal. Consequently, the high gravity factor in the current study was varied from 30 to 90 to maintain the nitrogen removal efficiency above 97%. The results are reasonable and acceptable. Here are some comments which should be carefully issued.

Comments:

1.       Figure 5, 6, 7, 8 was written as Figure 1, 2, 3, 4 again.

2.       The ANN models in this paper is lack of further experimental validation.

3.       Compared with other models, the author need to claim and prove the merits of this model in modeling Nitrogen Removal from Natural Gas.

The English language is fine

Author Response

(The authors gave the same response as above.)

Reviewer 3 Report

The paper by Surmi et al. presents ML/ANN-based models for nitrogen removal from natural gas. These results are extremely good for most of the models. The manuscript is very clear and the main conclusions are supported by the computational results. I  suggest acceptance with minor comments.

Figure 1 quality is really bad. Need to have bigger text.

It is always to have scripts/models and test/validation files in the SI in order for readers to reproduce some of the results.

It is okay.

Author Response

(The authors gave the same response as above.)

Round 2

Reviewer 1 Report

It can be accepted as it is.